# Soil responses to manipulated precipitation changes: An assessment of meta-analyses

Akane O. Abbasi[1], Alejandro Salazar[2,3], Youmi Oh[4], Sabine Reinsch[5], Maria del Rosario Uribe[1], Jianghanyang Li[4], Irfan Rashid[6], Jeffrey S. Dukes[1,2]

[1]Department of Forestry and Natural Resources, Purdue University, West Lafayette, 47907, USA
[2]Department of Biological Sciences, Purdue University, West Lafayette, 47907, USA
[3]Programa de Ciencias Básicas de la Biodiversidad, Instituto de Investigación de Recursos Biológicos Alexander von Humbold, Bogotá, 110311, Colombia
[4]Department of Earth, Atmospheric, and Planetary Sciences, Purdue University, West Lafayette, 47907, USA
[5]UK Centre for Ecology & Hydrology, Bangor, LL57 4TT UK
[6]Department of Botany, University of Kashmir, Srinagar, 190006, India

*Correspondence to*: Akane O. Abbasi (aota@purdue.edu)

**Abstract.** In the face of ongoing and projected climatic changes, precipitation manipulation experiments (PMEs) have produced a wealth of data about the effects of precipitation changes on soils. In response, researchers have undertaken a number of synthetic efforts. Several meta-analyses have been conducted, each revealing new aspects of soil responses to precipitation changes. Here, we conducted a comparative analysis of the findings of 16 meta-analyses focused on the effects of precipitation changes on 42 soil response variables, covering a wide range of soil processes. We examine responses of individual variables as well as more integrative responses of carbon and nitrogen cycles. We find strong agreement among meta-analyses that belowground carbon and nitrogen cycling accelerate under increased precipitation and slow under decreased precipitation while bacterial and fungal communities are relatively resistant to decreased precipitation. Much attention has been paid to fluxes and pools in carbon, nitrogen, and phosphorus cycles, such as gas emissions, soil carbon, soil phosphorus, extractable nitrogen ions, and biomass. The rates of processes underlying these variables (e.g., mineralization, fixation, and de/nitrification) are less frequently covered in meta-analytic studies, with the major exception of respiration rates. Shifting scientific attention to these less broadly evaluated "processes" would deepen the current understanding of the effects of precipitation changes on soil and provide new insights. By jointly evaluating meta-analyses focused on a wide range of variables, we provide here a holistic view of soil responses to changes in precipitation.

## 1 Introduction

Soil is an important component of terrestrial ecosystems through which carbon, nitrogen, phosphorus, and other elements cycle. Biological processes in soils, such as those driven by plant roots, microbes, and enzymes, regulate nutrient cycling, with direct impacts on aboveground plant and animal communities (Bardgett et al., 2008). Rates of biological activity in soils are largely determined by physical parameters, one of the most influential being soil moisture (Stark and Firestone,

1995; Brockett et al., 2012; Schimel, 2018). Historical observations have shown that annual precipitation has either increased or decreased significantly in many regions, and the intensity and frequency of precipitation extremes (heavy rainfalls and droughts) have likewise increased in many regions (Frei et al., 2006; Lenderink and van Meijgaard, 2008). These changes in
precipitation patterns are projected to continue in the future, possibly at a faster rate (Bao et al., 2017).

The activity of plant roots, microorganisms and enzymes is maximized at optimal soil water content, which is unique to each group of organisms, soil type and ecosystem (Bouwman, 1998; Schimel, 2018). Water in soil functions as (1) a resource to promote metabolism of microbes and plants, (2) a solvent of nutrients, and (3) a transport medium to provide pathways to
solutes and microorganisms (Schimel, 2018; Tecon and Or, 2017). In a water-limited environment, reduced belowground activities are common (Borken et al., 2006; Sardans and Peñuelas, 2005). The negative responses of soil processes to decreased precipitation are attributed to reduced metabolism of the organisms (Salazar-Villegas et al., 2016; Schimel et al., 2007), limited substrate availability/diffusivity (Manzoni et al., 2016), restricted mobility of the organisms (Manzoni et al., 2016), or a combination of these (Schimel, 2018). Increased precipitation, on the other hand, generally promotes processes
by shifting the soil moisture level closer to the optimum (Zhang et al., 2013; Zhou et al., 2013). However, excess water in soil often results in lower biological activity due to the limitation of oxygen flow (Bouwman, 1998; Reinsch et al., 2017), while anaerobic processes such as methane production are greatly promoted (Le Mer and Roger, 2001).

Natural variation in precipitation provides opportunities to observe responses of belowground activities (e.g., Goldstein et
al., 2000; Granier et al., 2007), but targeted studies of belowground responses are difficult. Controlled precipitation manipulation experiments offer the opportunity to specifically study ecosystem responses to changes in precipitation compared to naturally occurring fluctuations and have become common in recent decades (Beier et al., 2012; Borken et al., 2006; Knapp et al., 2017). Precipitation manipulation experiments (PMEs) involve constructing an experimental structure in the field, such as rainout shelters, curtains, and/or sprinklers, to simulate alternative precipitation patterns (Beier et al., 2012).
These setups enable direct comparisons between a manipulated precipitation treatment and a control (ambient precipitation) in the same study system, while keeping other environmental conditions nearly identical. PMEs have been established across ecosystem types and characteristics (biome, ecosystem, soil type, and land type), and often use different methodological approaches (e.g., in terms of the magnitude and duration of the precipitation manipulation, size of the experiment, method of rain exclusion, and/or variables measured) (Vicca et al., 2014).

A number of meta-analyses have assembled and synthesized large and diverse PME datasets (Blankinship et al., 2011; Canarini et al., 2017; Wu et al., 2011). The first to examine soil responses to precipitation changes was conducted by Wu et al. (2011), compiling 85 manipulation studies and presenting the changes in aboveground and belowground carbon dynamics. Since then, several additional meta-analyses have considered belowground responses to precipitation changes. As
of April 2019, according to our search criteria (details below), a total of 16 meta-analyses in this area were published. These

meta-analyses focused on different but complementary soil properties [e.g., soil C (Zhou et al., 2016) or N (Yue et al., 2019)]. A combined analysis of these meta-analyses would provide a holistic view of the potential effects of projected precipitation changes on soil processes.

In this paper, we conduct a comparative analysis of 16 meta-analyses that have examined soil responses to manipulated (increased and decreased) precipitation *in-situ*, encompassing 42 response variables including greenhouse gas exchanges, carbon and nitrogen dynamics, phosphorus content, microbial community, and enzyme activities. By collating the results of the published meta-analyses, we aimed to (1) provide a more holistic view of the effects of precipitation changes on soil composition and functioning, (2) discuss the potential underlying mechanisms of each response, and (3) identify knowledge

gaps and propose future research directions. This study covers an unusually wide range of soil processes and examines the responses of individual variables as well as nutrient cycles.

## 2 Review of meta-analyses

### 2.1 Meta-analysis collection

We collected peer-reviewed meta-analyses focused on the effects of decreased and/or increased precipitation on soil

variables. We collected meta-analyses that included only field studies where the magnitude of precipitation was manipulated. Some meta-analyses included both field and lab/greenhouse experiments, but we only analyzed field data in our comparisons. We used Google Scholar and Web of Science with the search terms "meta-analysis" AND "soil" AND ("respiration" OR "$CO_2$" OR "carbon" OR "nutrient" OR "nitro" OR "phosph" OR "$N_2O$" OR "$CH_4$" OR "microb" OR "enzyme" OR "bacteria" OR "fungi") AND ("altered precipitation" OR "drought" OR "decreased precipitation" OR

"increased precipitation" OR "water addition" OR "water reduction"). We identified 16 meta-analyses (Table 1); four of them focused on decreased precipitation (DP), one of them on increased precipitation (IP), and 11 on both DP and IP. A total of 42 soil variables were covered, encompassing a wide range of soil characteristics such as soil greenhouse gas exchanges, soil carbon, nitrogen, phosphorous, microbial and bacterial communities, enzymes, and physical characteristics of soil (Table 2). Only meta-analyses written in English and published before April 2019 were included in our analysis. All of the meta-

analyses except for Brzostek et al. (2012) collected observations globally, with a greater concentration of data in the United States (US), Europe, and China than other parts of the world. The dataset of Brzostek et al. (2012) is US-only, yet their data covers a wide range of ecosystem types and biomes.

### 2.2 Effect sizes

From each meta-analysis, we obtained the mean effect size of each soil variable. In this review, effect sizes are the natural

log of response ratios (lnRR) defined as:

$$lnRR = \ln\left(\frac{X_t}{X_c}\right),$$ (1)

where $X_t$ and $X_c$ are the mean values of the treatment (DP or IP) and control, respectively, for each observation. Homyak et al. (2017) used Hedge's $d$ instead of Eq. 1 for $N_2O$ emissions and N supply due to the negativity of RR. Hedge's $d$ is defined as $J(X_t-X_c)/S$ where $S$ is the pooled standard deviation, and $J$ is the correction of small sample bias (Homyak et al., 2017). Both lnRR and Hedge's $d$ are negative for inhibitory effects, and positive for stimulatory effects (Brzostek et al., 2012; Homyak et al., 2017). All meta-analyses calculated mean effect sizes and 95% confidence intervals (CI) with sample size or the inverse of the variance as the weighting function. The effect is considered significant when 95% CI does not overlap zero. Some meta-analyses applied additional weighting functions or normalized the measurements under different manipulation levels (Liu et al., 2016; Wu et al., 2011). We used these sample size- or variance-weighted effect sizes when available. We obtained the values from the main texts or supplementary materials of the articles. If necessary, we used the digitizing software Plot Digitizer (Huwaldt, 2015), to extract values from graphs. When only percent changes were reported, we converted to lnRR as in Ren et al. (2017, 2018):

$$lnRR = \ln\left(\frac{\% \; change}{100} + 1\right).$$ (2)

Some 95% CI were unavailable because points were not visible on graphs or because values of percent change below -100% were not convertible using Eq. 2 (e.g. He and Dijkstra, 2014). We also obtained the sample size, defined as the number of studies or observations included in the meta-analyses. The collected information is available in Abbasi et al. (2020).

## 3 Soil responses to precipitation changes

### 3.1 Responses of soil respiration and belowground biomass

Meta-analyses on autotrophic ($R_a$), heterotrophic ($R_h$), and total soil ($R_s = R_a + R_h$) respiration provide strong agreement that DP decreases, and IP increases, $R_s$, $R_a$, and $R_h$ (Fig. 1a). Litter biomass (B) follows the same pattern (Fig. 1b). Although the response of $R_a$ reaches significance in only one of two meta-analyses, the direction of the response is consistent. Responses of soil carbon variables [total carbon (C), soil organic C (SOC), and dissolved organic C (DOC)] to precipitation differ among meta-analyses, both in direction and significance (Fig. 1b). Interestingly, root B is strongly suppressed by both DP and IP. In contrast, IP stimulates belowground B and belowground net primary productivity (NPP), and DP increases root C (Fig. 1b). It is difficult to reconcile that IP suppresses root B but increases belowground B; the difference between the two measures is that belowground B includes not just roots, but also any other plant or animal-derived materials found in a soil core. We note that these two contrasting results come from different, single, meta-analyses with small sample sizes.

To understand the effects of precipitation on $R_s$, we need to understand the responses of roots, microbes, and substrates to DP and IP. When soil moisture is below field capacity and plants are active, $R_a$ and $R_h$, and belowground NPP are typically positively correlated with soil water availability. $R_a$ decreases under limited water supply due to (1) reduced plant growth

and nutrient demand, (2) reduced root tissue activity due to limited soil water, and (3) reduced respiratory substrate production from photosynthetic activity (Hasibeder et al., 2015). In contrast, increased water supply increases $R_a$ by enhancing plant growth and photosynthetic rates (Heisler-White et al., 2008; Maire et al., 2015). In concordance with these plant physiological responses, belowground NPP decreases with DP and increases with IP (Figure 1; Zhou et al., 2016). Belowground B also increases with IP. However, not all belowground responses follow this pattern; total C (which is also affected by microbial activity) increases with DP, and root B – with a very small sample size - decreases with IP (Fig. 1b).

Some responses vary by biome. For example, the effect of DP on total C is negative in temperate forests, and positive in tropical forests and grassland (Yuan et al., 2017; Zhou et al., 2016). Total C reflects a balance of plant inputs and microbial outputs, so differences in responses among systems may reflect differences in the strength of PME effects on plants vs. microbes across those systems. Responses of this metric also depend on the size of the initial pool relative to fluxes, and so may be differentially dampened across systems.

Responses of $R_a$ to DP and IP were either significant (Zhou et al., 2016) or non-significant (Liu et al., 2016), depending on the study (although the mean responses were consistent in direction across studies). The difference in significance could be attributed to small samples sizes and high variability in the case of DP. the samples sizes are somewhat larger for IP effects on $R_a$, and these responses depend on biome and $R_a$ separation method. For instance, significant IP effects were found in temperate forest and grassland, but not in boreal forest (Zhou et al., 2016), and $R_a$ separated from $R_h$ by clipping methods responded more positively than when trenching methods were used (Liu et al., 2016). Nonetheless, sample sizes remain relatively small, suggesting that additional research could help to identify how this process response varies with biomes and methods.

$R_h$ is the consequence of soil microbial activity decomposing soil organic matter (SOM) under aerobic conditions. SOM is frequently estimated by measuring its carbon component, SOC. $R_h$ is mainly regulated by microbial access to substrate and the physiological condition of microbes (Schimel, 2018). In dry soil, substrate tends to be isolated from microbes as solute mobility is low (Manzoni et al., 2012; Schimel, 2018). Furthermore, a great number of empirical observations and synthetic studies have shown that microbial activity is lower during droughts (Hueso et al., 2012; Jensen et al., 2003; Manzoni et al., 2012). This is because dry conditions force microbes into dormancy or shift their efforts from growth to survival (Salazar et al., 2018; Schimel et al., 2007). Wetting of dry soil, on the other hand, increases substrate availability to microbes (Skopp et al., 1990), makes microbes dispose of osmolytes from their body cells to regulate osmotic pressure (Schimel et al., 2007), and can activate dormant microbes (Salazar et al., 2018). These responses can be particularly rapid and strong, yielding pulses of respiration that are large enough to affect the net carbon exchanges in terrestrial ecosystems (Placella et al., 2012).

As with $R_a$, $R_h$ typically decreases under DP and increases under IP, with variations among biomes and $R_h$ separation methods. DP effects on $R_h$ are significant in boreal forest and wetland, but not in tropical and temperate forests (Zhou et al., 2016). Likewise, IP effects on $R_h$ are significant in forest and grassland, but not in wetland (Liu et al., 2016; Zhou et al., 2016). We hesitate to draw strong conclusions from these differences because of the relatively small sample sizes. Zhou et al. (2016), for example, have a sample size of four and five for the tropical and temperate forests, respectively, for DP, and

the effects are highly uncertain. The biomes with significant effects – wetlands under DP and grasslands under IP – have higher sample sizes, of 10 and 15, respectively. Biological mechanisms behind these differences can also be hypothesized, such as differences in microbial sensitivity to moisture across systems. Furthermore, the effects of DP and IP on soil respiration can depend on methodological factors of the field experiments not explicitly considered in all meta-analyses. For example, the effects of IP on $R_a$ can be significant when field work included clipping, but not when it included trenching

(Liu et al., 2016).

Overall, responses of $R_s$, $R_a$, and $R_h$ are positively correlated with precipitation changes and soil moisture (Liu et al., 2016; Ren et al., 2017; Zhou et al., 2016). Responses of SOC, DOC, and belowground NPP also tend to be positively correlated with precipitation changes (Ren et al., 2017; Zhou et al., 2016). Despite the broad agreement among meta-analyses, the

175 responses of respiration and soil carbon vary across studies, and can depend on biome, measurement method, treatment intensity, and other factors.

Microbial activity in soils is strongly controlled by the actions of enzymes (Ren et al., 2017). Many of these enzymes, which are produced and released by microbes, depolymerize complex carbon compounds (Ren et al., 2017). While enzyme activity

is relatively unresponsive to IP (Fig. 2), DP increases hydrolytic enzyme activity (breakdown of labile carbon) and inhibits oxidative activity (de-polymerization of recalcitrant carbon) (Fig. 2). This indicates that under dry conditions, the relative contributions of substrates from labile carbon sources increase, while the respective relative contributions from recalcitrant sources decrease.

The summary diagrams (Fig. 1c, 1d) illustrates how DP generally slows the belowground carbon cycle, while IP promotes it. Nearly all steps of the carbon cycle - carbon stock, substrates, microbial activity, and respiration – are altered by both types of precipitation changes. However, enzyme activity tends to be relatively unresponsive, particularly to IP, and the observations of biomass and carbon variables vary both in direction and significance among meta-analyses. These variables also tend to vary across biomes, ecosystems, and soil types.

**3.2 Responses of methane uptake**

We found only one meta-analysis that addressed the effects of precipitation on soil $CH_4$ (Yan et al., 2018). The results show a significant increase and decrease of soil $CH_4$ uptake in response to DP and IP, respectively (Fig. 1a). Soil $CH_4$ fluxes

involve two groups of microbes: methanogens and methanotrophs. Methanogens produce $CH_4$ and are predominantly active in anaerobic conditions, while methanotrophs oxidize $CH_4$ and are active in aerobic environments (Conrad, 2007). $CH_4$ oxidation seems to peak at 10-15% volumetric water content because these conditions favor methanotroph activity as well as $CH_4$ and $O_2$ diffusion (Adamsen and King, 1993; Del Grosso et al., 2000).

The results of Yan et al. (2018) were significant across a wide range of ecosystem types, treatment durations, and magnitudes of precipitation manipulation. The effects of DP were greater in farmlands than other land types, in shorter-term (< 1 year) experiments than longer-term ones, and in more extreme experiments (> 50% rain reduction). The effects of IP were greatest in boreal forest and in longer-term experiments (1-5 years) with greater rain addition (> 50%). However, a few empirical studies have shown opposite responses to this meta-analysis (Billings et al., 2000; Christiansen et al., 2015); for instance, a precipitation removal experiment in a floodplain decreased $CH_4$ uptake, possibly due to the acclimation of methanotrophs to high soil moisture conditions (Billings et al., 2000), or differences in the types of methanotrophs in floodplain (low-affinity methanotrophs) versus upland soil, where most $CH_4$ uptake occurs (Christiansen et al., 2015).

### 3.3 Responses of soil nitrogen dynamics

Several soil nitrogen variables, including root nitrogen (N), $N_2O$ emissions, total N, dissolved organic nitrogen (DON), and extractable $NH_4^+ + NO_3^-$ are significantly affected by precipitation changes (Fig. 3a). Specifically, DP decreases root N and $N_2O$ emissions and increases total N, DON, and extractable $NH_4^+ + NO_3^-$. We also found that two meta-analyses (sample sizes < 20) suggest no change in total N, while one (sample size = 156) suggests an increase with DP. Similarly, one meta-analysis suggests an increase of extractable $NH_4^+$ with DP while other two meta-analyses suggest no effects. In contrast, IP increases root N, $N_2O$ emissions, and extractable $NH_4^+$ (Fig. 3a). Two meta-analyses suggest that total N decreases with IP, while one meta-analysis suggests no effects.

Mineralization rate, defined as N supply by Homyak et al. (2017), does not change under DP despite the increase in substrate (i.e., DON) (Fig. 3). However, the product of mineralization and $N_2$ fixation is $NH_4^+$, which increases under DP according to one of three meta-analyses (Homyak et al., 2017) even though fixation could be suppressed (Hume et al., 1976; Streeter, 2003). This is reasonable considering that the consumption of $NH_4^+$ is likely to decrease with DP, mainly because of reduced plant nitrogen uptake (He and Dijkstra, 2014; Matías et al., 2011; Yuan et al., 2017) and microbial nitrogen assimilation (Homyak et al., 2017; Månsson et al., 2014). Homyak et al. (2017) found the increase in extractable $NH_4^+$ is greater under more intense DP. Nitrification and denitrification are expected to slow down with DP (Bouwman, 1998; Lennon et al., 2012; Stark and Firestone, 1995), also reducing $N_2O$ emission (Fig. 3b). This suggests that soil moisture could be a stronger regulator of nitrification and denitrification processes than the availability of $NH_4^+$ and $NO_3^-$ (Weier et al., 1993). The input (nitrification) and outputs (denitrification, plant uptake and microbial assimilation) of $NO_3^-$ both decline under DP, leaving extractable $NO_3^-$ unchanged (Fig. 3b).

Extracellular enzyme activity, here shown both as total proteolytic activity (pro-enzyme) and three particular N-acquisition enzyme activities (β-1,4-N-acetyl-glucosaminidase, leucine amino peptidase, and urease), does not change with DP or IP (Fig. 2). This indicates that the production of N-enzymes is not sensitive to water stress. Important outputs of the soil nitrogen cycle (denitrification and plant uptake) decrease while inputs remain constant or decline (Fig. 3b). As a result, total soil N increases or remains unchanged.

In contrast to DP, soil nitrogen cycling is accelerated by IP (Fig. 3c). Although no mineralization indicator was included in the meta-analyses, ample evidence shows that nitrogen mineralization is likely to increase with IP (Hu et al., 2014; Sierra, 1997; Pilbeam et al., 1993; Mazzarino et al., 1998). Along with greater $N_2$ fixation (Hume et al., 1976), which contributes to increasing $NH_4^+$ (Fig. 3c), positive responses are also expected in nitrification and denitrification rates (Bouwman, 1998; Niboyet et al., 2011; Stark and Firestone, 1995), plant nitrogen uptake (Schaeffer et al., 2013; Liu et al., 2016; Ma et al., 2013), and microbial nitrogen assimilation (Månsson et al., 2014), which result in increased $N_2O$ emissions, and lead to unchanged $NO_3^-$ as well as total N.

Soil nitrogen undergoes a wide range of chemical and biological transformations, some of which are difficult to quantify. Despite the large number of empirical studies included in meta-analyses, some nitrogen variables, such as rates of mineralization (for IP), nitrification, denitrification, and $N_2$ fixation, have not yet been examined in meta-analyses focused on PMEs.

**3.4 Responses of soil phosphorus**

We found four meta-analyses that examined how precipitation changes affect the soil phosphorus (P) cycle (He and Dijkstra, 2014; Yan et al., 2018; Yuan et al., 2017; Yue et al., 2018). The results differ among meta-analyses; for instance, according to these meta-analyses, IP can have a negative, positive, or non-significant effects on total P (Fig. 4). Yuan et al. (2017) assembled the largest dataset and found that IP decreases total P, while DP increases total P. As phosphorus is commonly a limiting nutrient for vegetation, plant P uptake and concentration are frequently studied, but studies of soil phosphorus stocks are rarer (He and Dijkstra, 2014; Yue et al., 2018). The timescale of precipitation experiments can be as short as one growing season (or less), and the effect of such short-term precipitation manipulations on slow processes such as chemical weathering is negligible. However, phosphorus cycling through faster processes such as decomposition of organic matter, plant uptake, and consumption by microbes can respond (Van Meeteren et al., 2007). Plant P uptake tracks in the same direction as changes in precipitation (He and Dijkstra, 2014). The effects on total P are strongly linked to soil type (Yuan et al. 2017). Although Yuan et al. (2017) found significant effects of DP and IP on total P, the effects were small (-0.1 < effect sizes < 0.1), and other meta-analyses show that soil P, as well as P-acquisition enzyme activity, are relatively unresponsive to

precipitation changes (Fig. 2, 4). Other global changes such as warming, elevated $CO_2$, and anthropogenic P and N deposition tend to have stronger impacts on the terrestrial P cycle than precipitation changes (Yue et al., 2018).

## 3.5 Responses of microbial biomass and community structure

Microbial biomass (MB) in soil either decreases or does not respond to DP (Fig. 5a), and these responses depend on the amount of precipitation removed (Zhou et al., 2016; Ren et al., 2017, 2018), the length of droughts (Ren et al., 2018), vegetation type (Zhou et al., 2016; Ren et al., 2017, 2018) and mean annual precipitation (MAP; Ren et al., 2017). MB is affected by DP only when precipitation is reduced by more than ~33% (Ren et al., 2017, 2018), the drought period is ≤ 2 years (Ren et al., 2018), and in wet (MAP > 600mm) regions (Ren et al., 2017). Additionally, vegetation type affects MB responses to DP; DP consistently decreases MB in forests (tropical and temperate but not in boreal; Zhou et al., 2016; Ren et al., 2017, 2018) and heathlands (Blankinship et al., 2011), but not in shrublands (Ren et al., 2017, 2018). A meta-analysis conducted by Zhou et al. (2016) found that DP decreases MB in grassland soils. However, more recent meta-analyses that included more studies (Ren et al., 2017, 2018) suggest that MB in grasslands does not respond to DP.

In contrast, IP stimulates microbial growth and thus increases MB unless the proportion added is very high (> +70%; Ren et al., 2017). Unlike DP, IP affects MB in dry (MAP < 600 mm) but not in wet (MAP > 600 mm) sites (Ren et al., 2017). This is consistent with IP increasing MB in soils from ecosystems that are generally water-stressed, such as deserts, shrublands, and grasslands (Zhou et al., 2016; Ren et al., 2017). Zhou et al. (2016) found that IP increases MB in soils in temperate forests. Other meta-analyses that included more studies (also including tropical forests) suggest that MB in forest soils is generally not affected by IP (Blankinship et al., 2011; Canarini et al., 2017; Ren et al., 2017). Overall, increased precipitation typically increases MB in the direct systems, where it makes conditions less extreme.

In contrast to the responsiveness of MB to altered precipitation, the composition of bacterial and fungal communities is rather unresponsive (Fig. 5b). Although Blankinship et al. (2011) and Yan et al. (2018) estimated significant effects on the abundance of fungi (both positive and negative effects of IP) and F:B ratio (negative effect of DP; n = 4), other studies with sample sizes an order of magnitude larger (e.g., Ren et al. 2018) estimated non-significant effects. The high resistance of bacteria and fungi to soil moisture changes has been frequently highlighted (Evans and Wallenstein, 2012; Schimel et al., 2007; Yuste et al., 2011). Fungi in particular, due to their filamentous structure, are capable of accessing substrates even in very dry soils (Manzoni et al., 2012). Bacteria and fungi also have a wide breadth of soil moisture niches; diverse types of bacteria and fungi tolerate water stress (Lennon et al., 2012). Differences in resistance between bacteria, fungi, and other functional types can alter microbial structure under precipitation changes; DP could promote a more fungi-dominated community (Yuste et al., 2011). Although gram-positive bacteria are more resistant to soil moisture changes than gram-negative bacteria due to their thicker and stronger cell walls (Schimel et al., 2007; Salazar et al., 2019), both gram-positive and negative bacteria have been unresponsive to DP (Fig. 5b). The sample sizes for bacteria and fungi in meta-analyses are

small compared to MB meta-analyses (Fig. 5). Although the currently available data cover a substantial range of locations and conditions, microbial responses within each site are likely to vary by treatment timing, intensity, frequency, and other environmental/climatic factors. Future studies of bacterial and fungal community responses can improve our understanding of the microbial responses to precipitation in terms of the composition and structure of the microbial community by more comprehensively exploring these factors.

## 3.6 Responses of belowground C:N:P stoichiometry

Belowground stoichiometric relationships of carbon, nitrogen, and phosphorus can help researchers interpret and infer nutrient movements in soil organisms and their environments. Yet, few meta-analyses have synthesized belowground stoichiometric responses to precipitation treatments; greater attention has been paid to stoichiometry of aquatic systems and plants (Cleveland and Liptzin, 2007; Redfield, 1958; Yuan and Chen, 2015). He and Dijkstra (2014) and Yan et al. (2018) found no changes in soil C:N and N:P with DP (Fig. 3), but MBC:MBN responded to both precipitation changes (Fig. 5). Increased MBC:MBN with IP indicates that wetter conditions stimulated greater metabolic activity of microbes, which accumulated more carbon in their bodies. This suggests that the soil microbial biomass C:N:P ratio, which is well-constrained globally (60:7:1) (Cleveland and Liptzin, 2007), could be altered by IP to have more weight on carbon. Soil N:P ratios can be heavily dependent on plant nutrient uptake; as discussed in Sect. 3.3, DP reduces plant nitrogen uptake, which could increase soil N:P. However, this effect depends on site aridity (Sardans et al., 2012), and could be mitigated by robust mycorrhizal symbioses (Mariotte et al., 2017), which could help maintain soil N:P ratios by sustaining plant nutrient uptake under DP.

## 4 Implications for future research

### 4.1 Knowledge gaps

Meta-analyses have substantially advanced our understanding of the impacts of precipitation changes on soil processes and properties. Responses of several variables have been investigated by three or more meta-analyses, and with robust datasets; these include soil respiration, nitrogen stocks, total phosphorus, and microbial biomass. However, many other variables have received less attention. For example, sample sizes for analyses of autotrophic respiration are smaller than heterotrophic respiration, substrate availability has not been analyzed while soil C, N and P content have, and analyses of bacterial and fungal responses to IP are sparser than DP. $CH_4$ fluxes have received less attention than $CO_2$ and $N_2O$, and no meta-analyses have examined the processes of nitrification, denitrification, and nitrogen fixation.

Filling these knowledge gaps could help to reveal the mechanisms underlying soil responses to precipitation changes. For example, there is robust agreement across studies that soil and heterotrophic respiration slow under DP and accelerate under IP. However, the relative importance of different mechanisms in the response of heterotrophic respiration is still unknown –

in other words, how much of this response comes from changes in the level of microbial activity (e.g., entering and exiting dormancy) vs. substrate availability? Similarly, what are the most important mechanisms behind changes in $N_2O$ emissions, and how quickly will total soil nitrogen respond? Interestingly, the variables receiving the greatest attention are largely the easier to measure "fluxes" (i.e., greenhouse gas emissions) and "pools" (i.e., soil carbon, biomass, and bacterial abundance).

Studies of processes that have received less attention (e.g., microbial metabolic state, nitrification, denitrification and N fixation) have the potential to inform models and improve predictions of the effects of precipitation changes on important fluxes and pools. This benefit can be seen in ecosystem models that explicitly represent active and dormant microbial biomass, which can outperform those representing microbial biomass as a single pool (He et al., 2015; Salazar et al., 2019; Wang et al., 2015). A more synthetic understanding of nitrification and denitrification responses across ecosystems could improve projections of societally relevant nitrate leaching and soil emissions of $N_2O$ and $NO_x$, and inform carbon-associated modeling, as the availability of N in ecosystems has a close connection with C sequestration (Barnard et al., 2005).

The meta-analyses we examined had strong geographical imbalances, as has been identified elsewhere. While all but one meta-analyses collected global empirical data, the data are concentrated in the US, Europe, and China. Almost 90% of the existing PMEs are located at mid-latitudes (30-60°), and there is an obvious sparsity at lower and higher latitudes (Beier et al., 2012). As a result, sample sizes for tropical and boreal ecosystems are substantially smaller than for temperate ecosystems in many of the meta-analyses. Studies of the effect of IP on $R_s$ provide good examples: Zhou et al. (2016) has a sample size of 13 for temperate forest, but only two for tropical forest and zero for boreal forest. Yan et al. (2018) features a larger sample size of 66 for temperate forest, but still has only four subtropical forest samples and two in boreal forest. The comprehensive meta-analysis recently conducted by Song et al. (2019) has similar geographical gaps. Expanding PMEs to the under-represented regions is critical in order to obtain a truly planetary synthesis.

**4.2 Challenges in meta-analyses and synthetic studies**

PMEs are quite diverse, adopting a variety of approaches, treatment levels, and treatment types (Beier et al., 2012; Kreyling and Beier, 2013), and so are the data derived from them. Many PMEs use long-term rainout shelters, which unavoidably modify the ambient environment in other ways (Kreyling et al., 2017). While synthesizing the results of PMEs around the globe in the context of these experimental issues could be challenging, meta-analyses provide one somewhat simplistic approach, through an exhaustive statistical summary of empirical studies (Hedges et al., 1999). Meta-analysis can obscure the substantial influence of environmental characteristics and methodological differences on effect sizes. Categorization by environmental characteristics, such as climate, geography, ecosystem, soil, and soil biota, can provide a local- to regional-view of soil responses that is specific to the given environmental characteristic. Categorization by methodology, such as experimental duration, intensity of treatment, measurement method, and fertilizer use, can clarify the human-derived impacts on effect sizes. These categorization efforts help to identify when and how soil responses depend on their environmental

context. While an exhaustive analysis of these categories is beyond the scope of this paper, we have highlighted the cases in which these factors affected each meta-analysis result in the text above. A further breakdown of these categories by environmental characteristics and methodology can be found in the Supplement (S1). As more and more PMEs are implemented, sample sizes available for meta-analysis are increasing (Song et al., 2019). In this regard, the recent deployment of broad networks of PMEs with standardized methodology and sampling procedures (Fraser et al., 2013; Halbritter et al., 2020) could ultimately contribute to more powerful meta-analyses with more easily interpreted outcomes (Hilton et al., 2019; Knapp et al., 2012, 2017).

We identified some technical challenges during this comparative study, including data collection and the definition of samples. Data collection is perhaps the most time-consuming process of searching literature and contacting researchers. Most meta-analyses extract effect size, standard deviation, and sample size from publication when possible, commonly with the use of digitizing software (Canarini et al., 2017; Liu et al., 2016; Ren et al., 2017; Xiao et al., 2018; Yan et al., 2018; and others). While digitizing software is helpful, the accuracy of the digitized values depends on the resolution of the figures. In some cases, digitizing is not feasible when points are too large, or error bars are too close to the points. Thus, we emphasize the importance of comprehensively presenting and publishing data, both in original studies and meta-analyses, to minimize errors associated with digitizing. Secondly, we found that the definition of a sample used in meta-analyses differs by studies. Specifically, some meta-analyses treat observations over multiple years from the same experiment as distinct individual samples, which could potentially violate the assumption of sample independence. We recommend, therefore, that a meta-analysis accounts for within-study dependency (Canarini et al., 2017) or selects a single year or season to include in the analysis. Lastly, we note that we aimed a comparison of existing meta-analyses to visualize the (in)consistency among the meta-analyses and identify the variables receiving more (or less) attention. We did not account for overlapping empirical data between meta-analyses, and thus, do not provide a unified dataset for new analyses. Instead, we clarified the sample sizes and publication year of each meta-analysis to help interpret the results.

**5 Conclusions**

This assessment of meta-analyses provides a broad perspective on how precipitation changes affect soils and belowground processes. Belowground carbon and nitrogen cycles speed up with increased precipitation and slow down with decreased precipitation, while bacterial and fungal communities are relatively insensitive to decreased precipitation. While the responses of the fluxes and pools of each cycle – gas emissions, soil carbon, nitrogen ions, and biomass – have been studied extensively, responses of the associated process rates remain less studied or unexamined by meta-analyses. There are also gaps in the study of soil elements such as phosphorus and nitrogen ions, as well as stoichiometric relationships, and bacterial/fungal biomass under increased precipitation. We suggest that additional scientific attention to these gaps is warranted, and would help to deepen and consolidate current knowledge of soil responses to precipitation changes.

**Data Availability**

The data collected from meta-analyses and used in this paper are available through the Purdue University Research Repository (https://doi.org/10.4231/16NT-CW47).

**Author contribution**

AOA, AS, YO, MRU, IR, and JSD designed the research. AOA, AS, YO, SR, MRU, JL, and IR conducted the comparative analysis and contributed to writing the original draft. AOA prepared the manuscript with contributions from all co-authors.

**Competing interests**

The authors declare that they have no conflict of interest.

**Acknowledgements**

Ideas for this paper were developed during a distributed graduate seminar organized by the Drought-Net Research Coordination Network (RCN) in spring 2017. Drought-Net was supported by the NSF (DEB-1354732; PIs: Melinda Smith, Osvaldo Sala, Richard Phillips). AOA was funded by the Department of Forestry and Natural Resources at Purdue University and Takenaka Scholarship Foundation in Tokyo, Japan. Most AS work was supported by funds from the Colciencias-Fulbright grant during his PhD at the Department of Biological Sciences at Purdue University; some work was supported by funds from the Icelandic Research Fund 2016, grant number 163336-052; and some from POA funds from the Instituto de Investigación de Recursos Biológicos Alexander von Humbold, Bogotá, Colombia. SR was supported by the Natural Environment Research Council award number NE/R016429/1 as part of the UK-SCAPE programme delivering National Capability. IR was funded by the University Grants Commission, India, under Raman Fellowship Programme. This is publication 2002 of the Purdue Climate Change Research Center.

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

**Table 1: List of meta-analyses used in this study.**

| No. | Meta-analysis |
| --- | --- |
| 1 | Blankinship et al. 2011 |
| 2 | Brzostek et al. 2012 |
| 3 | Canarini et al. 2017 |
| 4 | He & Dijkstra 2014 |
| 5 | Homyak et al. 2017 |
| 6 | Liu et al. 2016 |
| 7 | Ren et al. 2018 |
| 8 | Ren et al. 2017 |
| 9 | Wu et al. 2011 |
| 10 | Xiao et al. 2018 |
| 11 | Yan et al. 2018 |
| 12 | Yuan et al. 2017 |
| 13 | Yue et al. 2019 |
| 14 | Yue et al. 2018 |
| 15 | Zhou et al. 2016 |
| 16 | Zhou et al. 2018 |

Table 2. List of soil variables and their definitions as analyzed in the meta-analyses. The numbers indicate meta-analysis number corresponding to Table 1, examining the effects of decreased precipitation (DP) and increased precipitation (IP) on each soil variable.

| Variable | Definition | DP | IP | Variable | Definition | DP | IP |
|---|---|---|---|---|---|---|---|
| Rs | Soil respiration | 3,6,8,9,11,15 | 6,8,9,11,15 | $NH_4^+$ | Extractable $NH_4^+$ | 5,11,13 | 11 |
| Ra | Autotrophic respiration | 6,15 | 6,15 | $NO_3^-$ | Extractable $NO_3^-$ | 5,11,13 | 11 |
| Rh | Heterotrophic respiration | 6,8,15 | 6,8,15 | N:P | Extractable N:P | 4 | None |
| $CH_4$ | $CH_4$ uptake | 11 | 11 | Ext P | Extractable soil P | 4,14 | 14 |
| Total C | Total soil C | 11,12,15 | 11,12,15 | Total P | Total soil P | 11,12,14 | 11,12,14 |
| SOC | Soil organic C | 8 | 8 | MB | Microbial biomass | 3,5,7,8,16 | 1,8,16 |
| DOC | Dissolved organic C | 3,8,11 | 8,11 | MBC | Microbial biomass C | 11,15 | 10,11,15 |
| Litter B | Litter biomass | 11 | 11 | MBN | Microbial biomass N | 11,13 | 11,13 |
| Root B | Root biomass | 11 | 11 | MBC:MBN | Microbial biomass C: Microbial biomass N | 11 | 11 |
| Below B | Belowground biomass[a] | None | 9 | Bacteria | Abundance of bacteria | 7,11 | 1,11 |
| Below NPP | Belowground NPP | 15 | 9,15 | Fungi | Abundance of fungi | 7,11 | 1,11 |
| Root C | Fine root C concentration | 11 | 11 | Gram+ | Gram positive bacteria | 7 | None |
| Root N | Fine root N concentration | 11 | 11 | Gram- | Gram negative bacteria | 7 | None |
| Root C:N | Fine root C concentration: Fine root N concentration | 11 | 11 | F:B | Fungi:Bacteria ratio | 3,7,11 | 11 |
| C:N | Total soil C:N | 11 | None | Hy-enzyme — C-enzyme | Hydrolytic enzyme activity[b] — C-acquisition enzymes | | 10 | | 10 |
| $N_2O$ | $N_2O$ emissions | 5,11 | 11 | Hy-enzyme — N-enzyme | Hydrolytic enzyme activity[b] — N-acquisition enzymes | 8 | 10 | 8 | 10 |
| Total N | Total soil N | 11,12,13 | 11,12,13 | Hy-enzyme — P-enzyme | Hydrolytic enzyme activity[b] — P-acquisition enzymes | | 10 | | 10 |
| Inorganic N | Inorganic N | 13 | 13 | Ox-enzyme | Oxidase activity | 8,10 | 8,10 |
| N supply | N mineralization | 5 | None | Pro-enzyme | Potential proteolytic enzyme activity | 2 | 2 |
| DON | Dissolved organic N | 11 | None | Soil temperature | Soil temperature | None | 11 |
| $NH_4^+ + NO_3^-$ | Extractable $NH_4^+ + NO_3^-$ | 4 | None | pH | Soil pH | 11 | None |

a. Belowground biomass was measured by drying soil cores (Wu et al., 2011), and thus includes roots and other plant- and animal-derived materials. Root biomass includes biomass that derives from roots only.

b. C-acquisition enzymes are β-1,4-glucosidase and β-D-cellobiohydrolase, N-acquisition enzymes are β-1,4-N-acetyl-glucosaminnidase, leucine amino peptidase, and urease, and the P-acquisition enzyme is acid phosphatase (Xiao et al., 2018).

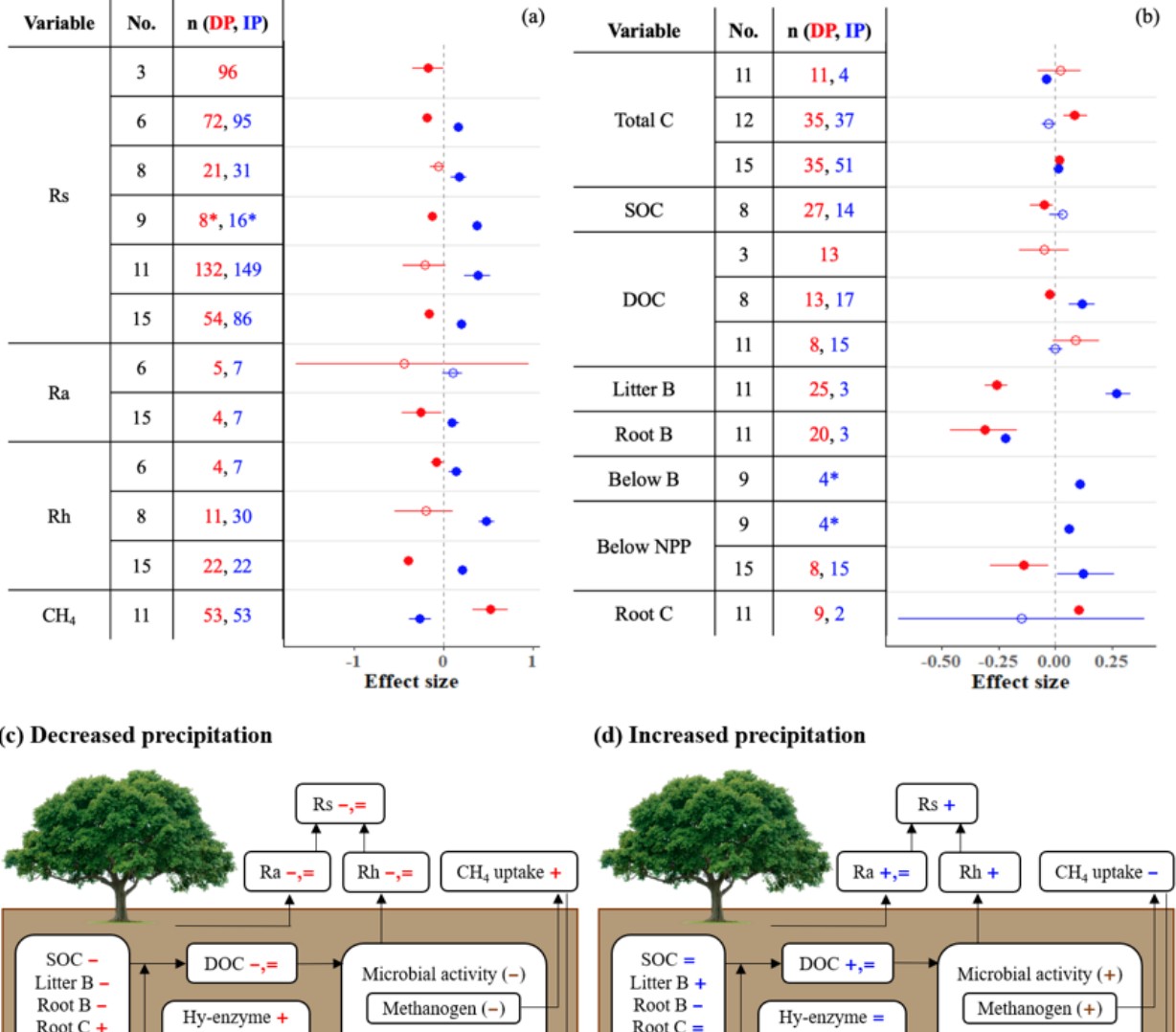

Figure 1: (a, b) Effect sizes for (a) soil respiration and methane uptake, and (b) carbon and belowground biomass variables with respect to decreased (red) and increased (blue) precipitation. Filled points represent significant effect sizes (95% CI not overlapping 0), and open points represent non-significant effect sizes. Variable names correspond to Table 2. No. is meta-analysis number as listed in Tables 1 and 2. The sample size is indicated by n. Asterisks indicate missing 95% CIs. (c, d) The effects of (c) decreased precipitation and (d) increased precipitation on the soil carbon cycle. Negative, positive, and non-significant effects are represented by −, +, and =, respectively. Red and blue represent variables found in one or more meta-analyses. Brown symbols in parentheses represent the variables that no meta-analyses quantified; in these cases, we estimated the effects based on our review of empirical studies in Sect. 3.1.

| Variable | No. | n (DP, IP) | |
|---|---|---|---|
| Hy-enzyme | 8 | 33, 52 | |
| C-enzyme | 10 | 16, 14 | |
| N-enzyme | 10 | 10, 10 | |
| P-enzyme | 10 | 9, 13 | |
| Ox-enzyme | 8 | 12, 13 | |
| | 10 | 5, 5 | |
| Pro-enzyme | 2 | 4, 4 | |
| Soil temperature | 11 | 13 | |
| pH | 11 | 10, 43 | |

Effect size

**Figure 2: Effect sizes for soil enzyme and physical variables with respect to decreased (red) and increased (blue) precipitation.** **Filled points represent a significant effect size (95% CI not overlapping 0), and open points represent a non-significant effect size. Variable names correspond to Table 2. No. is meta-analysis number as listed in Tables 1 and 2. The sample size is indicated by n.**

670

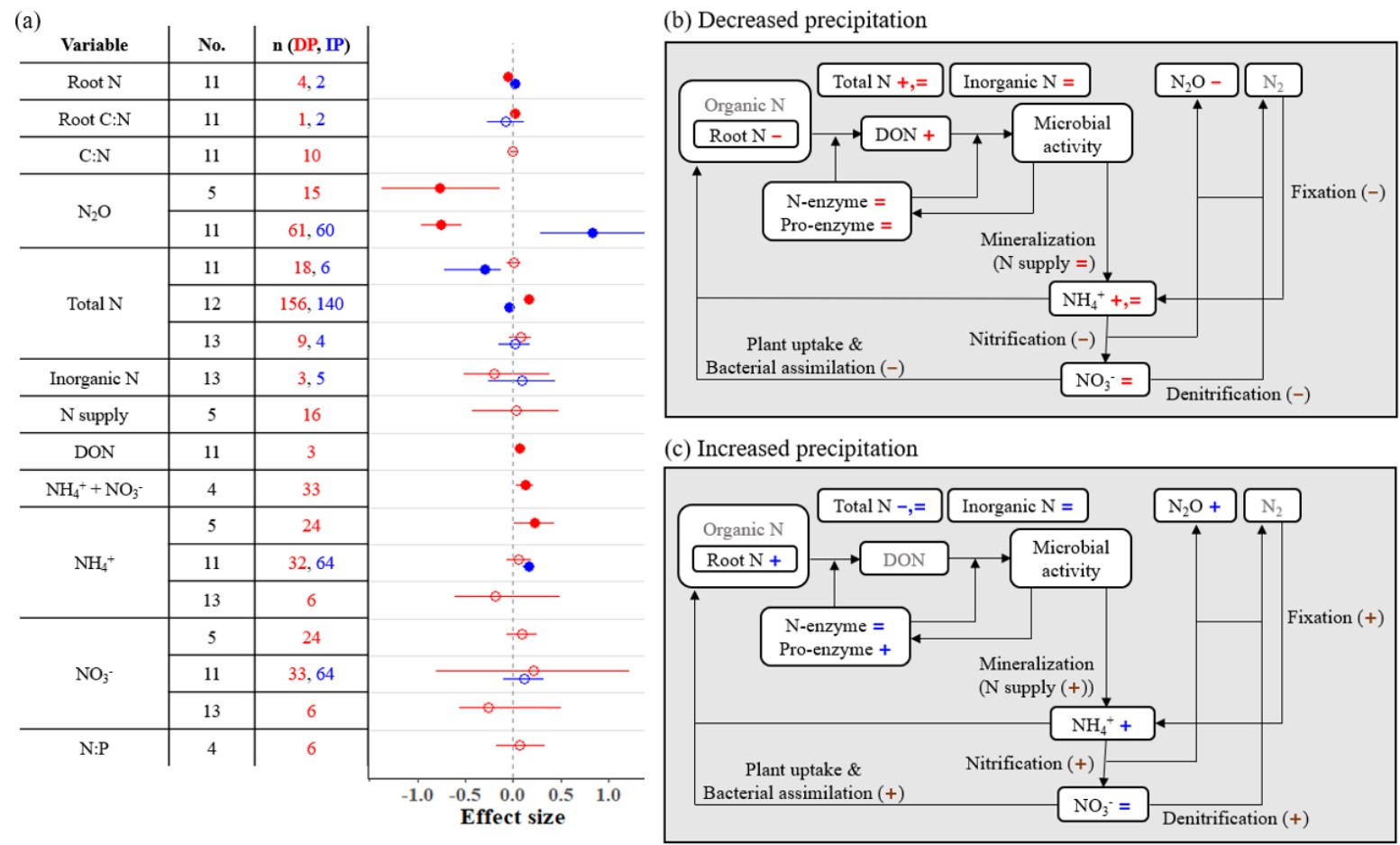

**Figure 3: (a) Effect sizes for soil nitrogen variables responding to decreased (red) and increased (blue) precipitation. Filled points represent a significant effect size (95% CI not overlapping 0), and open points represent a non-significant effect size. Variable names correspond to Table 2. No. is meta-analysis number as listed in Tables 1 and 2. The sample size is indicated by n. (b, c) The effects of (b) decreased precipitation and (c) increased precipitation on a simplified schematic of the soil nitrogen cycle. Negative, positive, and non-significant effects are represented by −, +, and =, respectively. These symbols are colored in red and blue if variables are found in one or more meta-analyses. Brown symbols in parentheses represent variables that no meta-analyses have quantified; in these cases, we estimated the effects based on our review of empirical studies in Sect. 3.3.**

| Variable | No. | n (DP, IP) |
|---|---|---|
| Ext P | 4 | 6 |
|  | 14 | 10, 3 |
| Total P | 11 | 1, 2 |
|  | 12 | 31, 36* |
|  | 14 | 6, 6 |

**Figure 4: Effect sizes for soil phosphorus variables responding to decreased (red) and increased (blue) precipitation. Filled points represent a significant effect size (95% CI not overlapping 0), and open points represent a non-significant effect size. Variable names correspond to Table 2. No. is meta-analysis number as listed in Tables 1 and 2. The sample size is indicated by n. Asterisks indicate missing 95% CIs.**

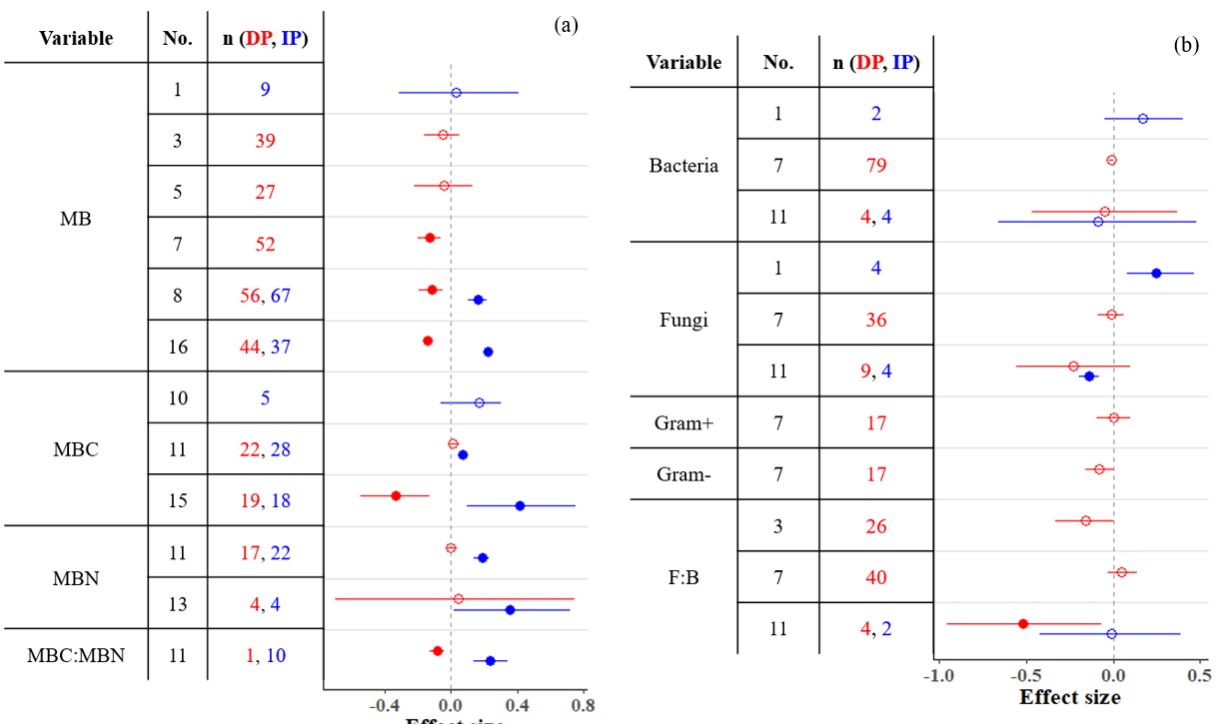

| Variable | No. | n (DP, IP) |
|---|---|---|
| MB | 1 | 9 |
|  | 3 | 39 |
|  | 5 | 27 |
|  | 7 | 52 |
|  | 8 | 56, 67 |
|  | 16 | 44, 37 |
| MBC | 10 | 5 |
|  | 11 | 22, 28 |
|  | 15 | 19, 18 |
| MBN | 11 | 17, 22 |
|  | 13 | 4, 4 |
| MBC:MBN | 11 | 1, 10 |

(a)

| Variable | No. | n (DP, IP) |
|---|---|---|
| Bacteria | 1 | 2 |
|  | 7 | 79 |
|  | 11 | 4, 4 |
| Fungi | 1 | 4 |
|  | 7 | 36 |
|  | 11 | 9, 4 |
| Gram+ | 7 | 17 |
| Gram- | 7 | 17 |
| F:B | 3 | 26 |
|  | 7 | 40 |
|  | 11 | 4, 2 |

(b)

**Figure 5: Effect sizes for (a) microbial biomass, carbon, and nitrogen, and (b) bacterial and fungal variables responding to decreased (red) and increased (blue) precipitation. Filled points represent significant effect sizes (95% CI not overlapping 0), and open points represent non-significant effect sizes. Variable names correspond to Table 2. No. is meta-analysis number as listed in Tables 1 and 2. The sample size is indicated by n.**