# Peer review of "Soil responses to manipulated precipitation changes: An assessment of meta-analyses"

_Biogeosciences, 2020_

## Referee Comment (RC1) · Feike Dijkstra (Referee) · 3 Mar 2020

In this paper Abbasi et al. summarises 16 meta-analyses about the effects of decreased and increased precipitation on 42 soil variables. This paper highlights the existence of a large number of meta-analyses, and highlights their consistencies and discrepancies, but unfortunately does not go much beyond that. Unlike what the title suggests, this is not a synthesis, but merely a summary, which is unfortunate. It provides some research gaps (e.g., lack of data on nitrification, denitrification and fixation), but even there, the authors do not really provide a rationale for WHY more information on this is needed. I also disagree about the statement in the abstract that "rates of processes underlying these variables are less frequently covered" than pools. Indeed, respiration rates (Figure 1) have some of the largest observations compared to some

of the pools.

I was further disappointed that no distinctions were made that go beyond effects of decreased and increased precipitation. It is well known that a large number of the 42 soil response variables listed here are quite dynamic in time and depend not only on the overall relative decrease or increase in precipitation, but also on timing, duration and frequency. I believe different soil responses to changes in precipitation among studies could for a large degree be described to differences in intensity and frequency, and I think this is a missed opportunity for discussing these issues in greater detail.

It was further unclear if only field studies were included when extracting the data from the 16 meta-analyses. I know some of the meta-analyses did include soil laboratory incubation studies, but I am not sure about all 16 meta-analyses. I can imagine that some of the soil variables would respond quite differently depending if they were measured in the field, greenhouse, or lab (and with or without plants).

Other comments: I was unclear what the difference was between "root biomass" and "belowground biomass" (Table 2). How are they different? l. 110: I guess strong agreement is not surprising if the same data are used for different meta-analyses. How much overlap in data used does there exist among the meta-analyses? l. 234: "humidity affects P deposition". How? I thought most atmospheric deposition of P was in the form of dry deposition. l. 268-272: I don't believe microbial community responses to precipitation changes are as clear as suggested here, and probably strongly depend on timing, intensity and frequency of the precipitation manipulation.
* * *

---

## Referee Comment (RC2) · Nameer Baker (Referee) · 16 Mar 2020

The authors present a useful meta-analysis of meta-analyses on the response of a wide variety of soil factors to increased or decreased precipitation. I believe that the authors have collated published data in a manner that merits publication, but I believe that the results of the study could be significantly improved if a consistent manner to combine and interpret data across meta-analyses could be employed, rather than the method of treating each meta-analysis as an individual unit for comparison. I appreciate that the authors do bring up the sample sizes of each meta-analysis when discussing them and weight the inferences drawn from larger studies more heavily, but I wonder if there is a more effective way to combine the results from the various studies to draw conclusions. Could meta-analyses that presented the same variables have the

effect sizes for that variable combined to produce one effect size for the response of the variable to changes in precipitation more generally across studies? This would require knowing the standard deviation of each variable within each meta-analysis, but would make for a much simpler presentation and interpretation of the data, as well as a more valid weighting of the results.

That is my main request that would require significant alterations to the text and the figures, but I do have some more easily implemented concerns, as well. I wonder if would it be possible to change the abbreviations "IP" and "DP" to something like "up arrow P" and "down arrow P," respectively. This would be easier for the reader to follow in the text, though you might then also want to think about using "W" instead of "P" to refer to precipitation/water to avoid then making it look as though phosphorus content is what is being discussed. Is there also a consistent way to talk about results that had a trend with precipitation, i.e. where something was reduced when precipitation was reduced and increased when precipitation was increased? For instance, saying a variable is positively correlated with precipitation across treatments or negatively correlated with precipitation across treatments? This might also be easier for readers to follow. Finally, I wonder if it also might make more sense to group enzyme results with microbial biomass, as they are a microbial response and that way you don't have to spread their discussion out over multiple sections. This is just a suggestion, however.

My specific comments are as follows:

94 – Did you also use Hedge's d for just these variables, or did you then use it for all variables?

122 – This is a place where it would help to be more explicit with your results given that you are saying they are in-line with an expectation, and you can use whatever way you decide to to talk about consistent trends with precipitation (for instance, the response of belowground NPP to both decreasing and increasing precip).

127 – If these are general trends across meta-analyses, then how do differences in

soil type explain these results more than the differences in nature of the C pool being measured?

149 – Are there any hypotheses as to why Rh is affected by decreased precip in boreal forest and wetlands, but not in tropical or temperate forests? How about for the effect of increased precipitation in forests and grasslands, but not in wetlands?

151 – It is unclear what the conclusion to be drawn from this sentence is.

185 – This is an example of where your presentation and discussion of results would benefit greatly from being able to combine effect sizes across meta-analyses for like variables.

192 – This is a difficult sentence to parse, I'm not sure how best to remedy it but perhaps something like "However, the product of mineralization and N2 fixation is NH4+, and it increases under DP according to one of three meta-analyses even though fixation could be suppressed."

225 – This brings up something that is a bit lacking in discussion of these meta-analyses – are any of them biased or targeted in some fashion, or are they all global? And would a geographic analysis of where all the study sites employed in all of the meta-analyses reveal some obvious blind spots or areas that have been over-represented in the literature? These would be valuable conclusions to be able to make as a result of your study.

237 – It may be worth bringing up timescale of studies here for reference relative to P-weathering rates.

255 – It seems that you may be able to draw the conclusion that moisture appears to be generally limiting for microbes in soil.

260 – What direction was this response?

271 – I am not sure that this is the best way to phrase this result, as it appears more

that changes in precip don't favor one over the other.

281 – In what direction could the ratio be altered?

283 – How would the mycorrhizal symbiosis change the dynamics? A bit more detail would be useful to the reader.

287-292 – This section feels sparse, and would be well-served to also bring up ecosystems or geographic regions that have been under- or over-sampled, as mentioned in a previous comment. Also, what about the paucity of studies that have measured bacterial:fungal biomass responses to increased precipitation?

301 – Some more discussion of what this blind spot in terms of N-process rates means for inferring conclusions about the N-cycle in soil would be useful to the reader to understand why this is valuable fruit to pursue.

317 – Do you have any suggestions as to what types of data formatting / archiving you ran into that was helpful or a hindrance? You have an opportunity to say some things from this pulpit, take advantage!

322 – I'm not sure this statement is quite true given the response of microbial biomass and the crude measures of microbial community assayed – it is fair to say that the ratio of fungi to bacterial biomass is insensitive, but that is not the same as the community being resistant.

Figures – Could you bold the symbols used to indicate the direction of the effect to make them stand out more? Also, if you are not going to use the raindrops to denote precip effects on each flow-figure then don't use it on any of them.

―――――――――――――――――――――

---

## Author Comment (AC1) · 18 Mar 2020

We would like to thank Dr. Dijkstra for providing his valuable feedback on our manuscript. Here is a point-by-point response to his comments and concerns.

Comment 1: Unlike what the title suggests, this is not a synthesis, but merely a summary, which is unfortunate. Response: We attempted to synthesize the meta-analyses in section 3, and also visualized the outcome in Figures 1 and 3, which could be considered synthetic. Based on your comment, we are considering adding a subsection 3.7 to more fully synthesize what we learned from section 3.1-3.6. We would welcome additional input on specific areas that could be made more synthetic.

Comment 2: It provides some research gaps (e.g., lack of data on nitrification, den-

itrification and fixation), but even there, the authors do not really provide a rationale for WHY more information on this is needed. Response: We believe that filling such research gaps will strengthen the current understanding of precipitation change effects. For example, we know that, in general, decreased precipitation decreases soil N2O emissions, and increased precipitation increases soil N2O emissions. However, we have little understanding of why these changes occur (i.e., because of increased/decreased nitrification, denitrification, mineralization, or combinations of those?). Clarified mechanisms underlying the soil N2O responses may help mitigation strategies and management practices. We are considering adding a discussion of the rationale in section 4.1.

Comment 3: I also disagree about the statement in the abstract that "rates of processes underlying these variables are less frequently covered" than pools. Indeed, respiration rates (Figure 1) have some of the largest observations compared to some of the pools. Response: Yes, respiration is the exception; it is one of the most frequently covered variables. We mentioned "rates of processes" generally in the abstract, specifically listed rates of mineralization, fixation, and de/nitrification as an example. We will clarify that there are some processes that are frequently studied, such as respiration, so there will no longer be confusion. Thank you for pointing this out.

Comment 4: I was further disappointed that no distinctions were made that go beyond effects of decreased and increased precipitation. It is well known that a large number of the 42 soil response variables listed here are quite dynamic in time and depend not only on the overall relative decrease or increase in precipitation, but also on timing, duration and frequency. I believe different soil responses to changes in precipitation among studies could for a large degree be described to differences in intensity and frequency, and I think this is a missed opportunity for discussing these issues in greater detail. Response: This is a great point, and we did initially attempt to look in detail at treatment timing, duration, intensity, and frequency, as well as other methodological and environmental factors. However, covering so many variables both for decreased and

increased precipitation already led to a lengthy manuscript, and we decided to simply discuss the effects of treatment timing and other factors briefly in the text. Therefore, in each section, we highlighted cases in which these factors affected each meta-analysis result. The supplemental information also discusses the importance of these factors in further detail, as well as how frequently they are taken into account in meta-analyses. We believe that another project could more fully incorporate methods and environmental differences. Still, our paper focused on the general effects of precipitation changes that include all variations of manipulation setups.

Comment 5: It was further unclear if only field studies were included when extracting the data from the 16 meta-analyses. I know some of the meta-analyses did include soil laboratory incubation studies,but I am not sure about all 16 meta-analyses. I can imagine that some of the soil variables would respond quite differently depending if they were measured in the field, greenhouse, or lab (and with or without plants). Response: Section 2.1. Specifies that "we collected meta-analyses that included only field studies where the magnitude of precipitation was manipulated." Some meta-analyses showed both field experiments and lab/greenhouse experiments, but we did not include the lab/greenhouse experiments' results. We will add a sentence to clarify this point.

Comment 6: I was unclear what the difference was between "root biomass" and "belowground biomass" (Table 2). How are they different? Response: Belowground biomass was measured by drying soil cores (Wu et al., 2011), and thus includes roots and other plant and animal materials. Root biomass, as the name suggests, includes biomass that derives from roots only. We will clarify this with a footnote in the table.

Comment 7: l. 110: I guess strong agreement is not surprising if the same data are used for different meta-analyses. How much overlap in data used does there exist among the meta-analyses? Response: This is a great question, and we have to note that we did not set out to do our own meta-analysis, and we did not specifically analyze the overlap in data across the meta-analyses we found. However, one can guess the extent of overlap from the sample size and study year. Taking soil respiration (Rs) as

an example, the sample size for decreased precipitation is 96, 72, 21, 8, 132, and 54 from six different meta-analyses. The studies with sample sizes of 132 and 8 probably have minimal overlap, while the study with 132 and 96 may contain substantial overlap. Also, the publication years of the meta-analyses range from 2011 to 2018, and newer studies likely to include data that earlier studies could not have included. We think it is important to show that, while every meta-analysis has a unique sample size and time range, there is typically strong agreement among them for any given variable.

Comment 8: l. 234: "humidity affects P deposition". How? I thought most atmospheric deposition of P was in the form of dry deposition. Response: While some P is deposited through dissolved P in rain, mist, and snow, the amount is typically quite small, and this phenomenon is not critical for this manuscript. We will delete this statement.

Comment 9: l. 268-272: I don't believe microbial community responses to precipitation changes are as clear as suggested here, and probably strongly depend on timing, intensity and frequency of the precipitation manipulation. Response: Thank you. We agree that these caveats are important. We plan to change this concluding remark of this section to highlight these dependencies.

---

## Author Comment (AC2) · 26 Mar 2020

We would like to thank Dr. Baker for providing his valuable feedback on our manuscript. Here is a point-by-point response to his comments and concerns.

Comment 1: The authors present a useful meta-analysis of meta-analyses on the response of a wide variety of soil factors to increased or decreased precipitation. I believe that the authors have collated published data in a manner that merits publication, but I believe that the results of the study could be significantly improved if a consistent manner to combine and interpret data across meta-analyses could be employed, rather than the method of treating each meta-analysis as an individual unit for comparison. I appreciate that the authors do bring up the sample sizes of each meta-analysis when

discussing them and weight the inferences drawn from larger studies more heavily, but I wonder if there is a more effective way to combine the results from the various studies to draw conclusions. Could meta-analyses that presented the same variables have the effect sizes for that variable combined to produce one effect size for the response of the variable to changes in precipitation more generally across studies? This would require knowing the standard deviation of each variable within each meta-analysis but would make for a much simpler presentation and interpretation of the data, as well as a more valid weighting of the results. Response: Thank you for your suggestion, and we agree that having one effect size for each variable would greatly simplify the presentation and interpretation. However, we find it challenging to implement because a few 95% CIs are missing, and there are some overlaps of empirical data used among meta-analyses, and deriving one effect size wouldn't be an accurate calculation. The fact that there are some overlaps of empirical data has been pointed out by the first referee, and for this reason, we included sample size and publication year of each meta-analysis (please see our complete response as part of the interactive discussion here: https://www.biogeosciences-discuss.net/bg-2020-30/). Furthermore, a few of the merits of showing individual meta-analyses include; it visualizes the (in)consistency among the meta-analyses in results, and it visualizes which variables have been more frequently covered compared to other variables.

Comment 2: That is my main request that would require significant alterations to the text and the figures, but I do have some more easily implemented concerns, as well. I wonder if would it be possible to change the abbreviations "IP" and "DP" to something like "up arrow P" and "down arrow P," respectively. This would be easier for the reader to follow in the text, though you might then also want to think about using "W" instead of "P" to refer to precipitation/water to avoid then making it look as though phosphorus content is what is being discussed. Response: Thank you for your suggestion. It is a great point that P might be a confusing letter for phosphorus. We are considering changing "IP" and "DP" to "up arrow W" and "down arrow W".

Comment 3: Is there also a consistent way to talk about results that had a trend with precipitation, i.e. where something was reduced when precipitation was reduced and increased when precipitation was increased? For instance, saying a variable is positively correlated with precipitation across treatments or negatively correlated with precipitation across treatments? This might also be easier for readers to follow. Response: Thank you for your suggestion. While we agree that showing a trend with precipitation could be useful information, it is difficult to make, for example, a graph of x = precipitation change (%) and y = effect size and show the relationship (= trend) because a meta-analytic result could include multiple precipitation change levels. We are, however, able to give a summary of the estimated trend from the meta-analyses in a consistent way, and we are considering revising our writing in that manner.

Comment 4: Finally, I wonder if it also might make more sense to group enzyme results with microbial biomass, as they are a microbial response and that way you don't have to spread their discussion out over multiple sections. This is just a suggestion, however. Response: We initially considered including enzymes in the microbial biomass section. However, as there are respective enzymes for carbon, nitrogen, and phosphorus cycles, we decided to break them into each section. In this way, we were able to summarize enzyme responses in each cycle and relate their responses to other components of the cycle.

Comment 5: 94 – Did you also use Hedge's d for just these variables, or did you then use it for all variables? Response: We used Hedge's d for only these variables.

Comment 6: 122 – This is a place where it would help to be more explicit with your results given that you are saying they are in-line with an expectation, and you can use whatever way you decide to to talk about consistent trends with precipitation (for instance, the response of belowground NPP to both decreasing and increasing precip). Response: Thank you for your suggestion. We are considering adding the trends with precipitation here.

Comment 7: 127 – If these are general trends across meta-analyses, then how do differences in soil type explain these results more than the differences in nature of the C pool being measured? Response: The impact of differences in soil type is just one of the possibilities. Because the meta-analyses suggest that soil type and biome could affect some of the variables' responses to precipitation changes, we show that these could be one of the reasons for the inconsistent evidence described in l. 124.

Comment 8: 149 – Are there any hypotheses as to why Rh is affected by decreased precip in boreal forest and wetlands, but not in tropical or temperate forests? How about for the effect of increased precipitation in forests and grasslands, but not in wetlands? Response: We believe it is primarily due to the small sample size. Zhou et al. (2016), for example, have a sample size of 4 and 5 for the tropical and temperate forests, respectively, for decreased precipitation, and the effect is highly uncertain given the small sample size. The biomes with significant effect - wetlands in decreased precipitation and grasslands in increased precipitation - are 10 and 15 in sample size, respectively. Biological hypotheses can also be drawn, such as differences in microbial sensitivity. We are considering adding a discussion here.

Comment 9: 151 – It is unclear what the conclusion to be drawn from this sentence is. Response: This paragraph introduces variability in effect size depending on biomes, methods, and other factors. We are considering adding a discussion suggested above and concluding that the general effect could be different depending on multiple factors.

Comment 10: 185 – This is an example of where your presentation and discussion of results would benefit greatly from being able to combine effect sizes across meta-analyses for like variables. Response: Thank you for specifying the point where we can improve the manuscript by combining effect sizes. As we described earlier, it is challenging for us to combine effect sizes. We, however, appreciate your suggestion and leave it for the next project to achieve.

Comment 11: 192 – This is a difficult sentence to parse, I'm not sure how best to

remedy it but perhaps something like "However, the product of mineralization and N2 fixation is NH4+, and it increases under DP according to one of three meta-analyses even though fixation could be suppressed." Response: Thank you for your suggestion. We are considering changing the sentence accordingly.

Comment 12: 225 – This brings up something that is a bit lacking in discussion of these metaanalyses – are any of them biased or targeted in some fashion, or are they all global? And would a geographic analysis of where all the study sites employed in all of the meta-analyses reveal some obvious blind spots or areas that have been overrepresented in the literature? These would be valuable conclusions to be able to make as a result of your study. Response: None of the meta-analyses has targeted region/country/biome to conduct their meta-analysis, meaning that they all include empirical observations from the world. Yet, the observations are concentrated in the US, Europe, and East Asia, and are sparse in other regions. We are considering adding a discussion regarding this point.

Comment 13: 237 – It may be worth bringing up timescale of studies here for reference relative to P-weathering rates. Response: Thank you for your suggestion. We are considering commenting on the time scale of the studies.

Comment 14: 255 – It seems that you may be able to draw the conclusion that moisture appears to be generally limiting for microbes in soil. Response: Thank you for your suggestion. Yes, it is important to have a concluding sentence here, and we are considering adding one as your suggestion.

Comment 15: 260 – What direction was this response? Response: The sentences you pointed to are showing a non-significant effect, so there is no direction. If you meant "Although Blankinship et al. (2011) and Yan et al. (2018) estimated significant effects on the abundance of fungi and F:B ratio (n = 4), ...", both negative and positive effects of IP on the abundance of fungi, and negative effect of DP on F:B ratio. We are considering the clarification of these effects.

Comment 16: 271 – I am not sure that this is the best way to phrase this result, as it appears more that changes in precip don't favor one over the other. Response: We had a similar comment from the first referee, and we are considering emphasizing the variability of effects based on the magnitude, duration, and timing of the precipitation treatment in this concluding sentence.

Comment 17: 281 – In what direction could the ratio be altered? Response: As MBC:MBN increased with IP, soil microbial biomass C:N:P could also be increased to have more weight on carbon. We are considering clarification of this point.

Comment 18: 283 – How would the mycorrhizal symbiosis change the dynamics? A bit more detail would be useful to the reader. Response: Strong mycorrhizal symbiosis might be able to help plant's nutrient uptake under DP and help maintain soil N:P ratio. We are considering adding more detailed descriptions.

Comment 19: 287-292 – This section feels sparse, and would be well-served to also bring up ecosystems or geographic regions that have been under- or over-sampled, as mentioned in a previous comment. Also, what about the paucity of studies that have measured bacterial:fungal biomass responses to increased precipitation? Response: Thank you for your suggestion. It is a great point to include the geographic differences in observations, as well as the paucity of studies in bacterial:fungal biomass responses. We are considering improving this paragraph based on your comment.

Comment 20: 301 – Some more discussion of what this blind spot in terms of N-process rates means for inferring conclusions about the N-cycle in soil would be useful to the reader to understand why this is valuable fruit to pursue. Response: We had a similar comment from the first referee as well, and we agree that we need to elaborate on the importance of these nitrogen process rates variables. We are considering improving the section by clarifying the values of these variables.

Comment 21: 317 – Do you have any suggestions as to what types of data formatting / archiving you ran into that was helpful or a hindrance? You have an opportunity to

say some things from this pulpit, take advantage! Response: Thank you for the great suggestion. We are considering adding a discussion on data formatting and archiving in this section.

Comment 22: 322 – I'm not sure this statement is quite true given the response of microbial biomass and the crude measures of microbial community assayed – it is fair to say that the ratio of fungi to bacterial biomass is insensitive, but that is not the same as the community being resistant. Response: Thank you for your guidance. We agree with your point, and we are considering revising the section based on your comment.

Comment 23: Figures – Could you bold the symbols used to indicate the direction of the effect to make them stand out more? Also, if you are not going to use the raindrops to denote precip effects on each flow-figure then don't use it on any of them. Response: Thank you for the suggestions. The symbols are actually already bold, but we are going to make them more stand out. And yes, we are going to remove the raindrops used in Figure 1.

---

## Author Comment (AC3) · 1 Apr 2020

We discussed Dr. Baker's comment #2 further, in which he suggests that we change "IP" and "DP" to "up arrow W" and "down arrow W" as "P" can be confusing with phosphorus. While we understand the reason for the suggestion, "W" can also cause confusion in the global change community as "W" often refers to warming. We also see IP and DP used in other literature (such as Zhou et al. 2018), and we believe that IP and DP would not cause significant confusion, especially that we clearly define IP and DP in the text and each figure's caption. We highly appreciate his suggestion, but we decided to keep IP and DP.

---

## Author Response (AR1)

Dear Dr Kees Jan van Groenigen,

Thank you for the opportunity to revise our manuscript and for your thoughtful comments. We appreciate the careful review and constructive suggestions by the referees that helped us improve our manuscript considerably. We hereby submit our revised manuscript entitled "Soil responses to manipulated precipitation changes: An assessment of meta-analyses." As indicated in the responses that follow, we have addressed all the comments made by you and the referees in the revised manuscript.

**Editor's comments**

- **Comment 1**: Your manuscript has now been seen by two reviewers. Both of them provided some excellent suggestions to improve your manuscript. While I fully agree with both reviewers that a quantitative synthesis of the various meta-analyses would be a worthwhile effort, this would amount to an enormous amount of extra work. It would require access to the raw data of all meta-analyses (to remove overlap between datasets; calculating an average across meta-analyses without removing such overlap would amount to pseudo-replication). About 10 years ago I was involved in such a "meta-meta-analysis" for just one variable (soil C stocks under elevated CO2), synthesising results for only 4 meta-analyses (see Hungate et al. 2009, GCB). That analysis alone resulted in an enormous amount of work; doing the same thing for 16 meta-analyses and 42 variables is not realistic.
  However, I agree with both reviewers that "synthesis" in the title implies that this is exactly the kind of work you would be doing. A proper synthesis would also address points like the one brought up in comment 4 by Dr. Dijkstra. Perhaps the authors could choose a phrase that more accurately describes their approach? "Comparison" or "assessment" could both work.

  **Response to Comment 1**: Thank you for your understanding regarding the difficulty of conducting a meta-analysis of meta-analyses. We also thought of conducting a meta-analysis at the very initial stage of this project. However, we realized that many meta-analyses have been already conducted on the same variables of interest, but in some cases, yielded contradicting results. This is how we came up with this idea of comparing multiple meta-analyses, and we believe that it is beneficial information to show to the community. We agree that "synthesis" could be misleading in the title and in the text, and therefore, we changed the title to "Soil responses to manipulated precipitation changes: An assessment of meta-analyses". We also replaced "synthesis" or "summary" with "comparison" or "assessment" in the text.

- **Comment 2**: Both reviewers provide good suggestions to add depth to your discussion, and you indicated you would be willing to incorporate these. In the absence of a true quantitative synthesis of the meta-analysis, addressing comment 2 by Dr. Dijkstra and comment 21 by Dr. Baker seem especially important.

  **Response to Comment 2**: Thank you for your specific advice. Please find our responses to each comment below.

**Referee #1: Feike Dijkstra**

- **Comment 1**: Unlike what the title suggests, this is not a synthesis, but merely a summary, which is unfortunate.

  **Response to Comment 1**: We see how the word "synthesis" in the title could give the wrong impression. We have now changed the title to "Soil responses to manipulated precipitation changes: An assessment of meta-analyses", and replaced "synthesis" in the text with "comparison" or "assessment". In this way, the audience should expect to read about what we actually did in this paper - not a meta-analysis of meta-analyses or a giant combined analysis but rather a comparative study. We would like to kindly note that conducting a combined synthesis that incorporated all original data from all studies was not realistic for us considering the time and effort required. Although our original idea was to conduct a true synthesis, we immediately realized that it was infeasible for us (also the recent large-scale meta-analysis by Song et al. (2019) took a similar role, if with a somewhat different focus), and chose instead to compare the existing meta-analyses. We found that a good number of meta-analyses had been already conducted on many of the same variables, but they sometimes yielded contradictory results. We believe that our comparative study, providing an overview across many studies, has value and is worth presenting to the scientific community.

- **Comment 2**: It provides some research gaps (e.g., lack of data on nitrification, denitrification and fixation), but even there, the authors do not really provide a rationale for WHY more information on this is needed.

  **Response to Comment 2**: We believe that these "process" variables need further examination because they have a greater potential of informing model design and helping to evaluate model responses. For example, Salazar et al. (2019) has shown that incorporating microbial metabolic state (active vs. dormant) could improve $R_h$ models compared to the models based solely on physical predictors. Meta-analytical treatments of processes such as nitrification and denitrification could reduce uncertainty related to the representation of these processes in models; accurate representation is important for projecting societally relevant changes in variables such as nitrate leaching and soil emissions of $N_2O$ and $NO_x$. We included this discussion in Sect. 4.1 (*l.* 349-356).

- **Comment 3**: I also disagree about the statement in the abstract that "rates of processes underlying these variables are less frequently covered" than pools. Indeed, respiration rates (Figure 1) have some of the largest observations compared to some of the pools.

  **Response to Comment 3**: Yes, this is a good point. Respiration is the most obvious exception; it is one of the most frequently covered variables. We mentioned "rates of processes" generally in the abstract, and specifically listed rates of mineralization, fixation, and de/nitrification as examples. We clarified that there are some processes that are frequently studied, such as respiration (*l.* 25-26), so there should no longer be confusion.

- **Comment 4**: I was further disappointed that no distinctions were made that go beyond effects of decreased and increased precipitation. It is well known that a large number of the 42 soil response

variables listed here are quite dynamic in time and depend not only on the overall relative decrease or increase in precipitation, but also on timing, duration and frequency. I believe different soil responses to changes in precipitation among studies could for a large degree be described to differences in intensity and frequency, and I think this is a missed opportunity for discussing these issues in greater detail.

**Response to Comment 4**: This is a great point, and we did initially attempt to cover in detail differences related to treatment timing, duration, intensity, and frequency, as well as other methodological and environmental factors. However, even covering the general responses to so many variables for both decreased and increased precipitation led to a lengthy manuscript, and we decided that the text should only discuss the effects of treatment timing and other factors briefly. Therefore, in each section, we concisely highlighted cases in which these factors affected each meta-analysis result. In the supplemental information, we discuss the importance of these factors in further detail, as well as how frequently they are taken into account in meta-analyses. We clarified this point in Sect. 4.2 (l. 378-381). We believe that another project could more fully examine methodological and environmental differences.

- **Comment 5**: It was further unclear if only field studies were included when extracting the data from the 16 meta-analyses. I know some of the meta-analyses did include soil laboratory incubation studies,but I am not sure about all 16 meta-analyses. I can imagine that some of the soil variables would respond quite differently depending if they were measured in the field, greenhouse, or lab (and with or without plants).

   **Response to Comment 5**: In section 2.1 we specify that "we collected meta-analyses that included only field studies where the magnitude of precipitation was manipulated." We have now added "Some meta-analyses included both field and lab/greenhouse experiments, but we only analyzed field data in our comparisons." to clarify this point (*l*. 84-85).

- **Comment 6**: I was unclear what the difference was between "root biomass" and "belowground biomass" (Table 2). How are they different?

   **Response to Comment 6**: Belowground biomass was measured by drying soil cores (Wu et al., 2011), and thus includes roots and other plant and animal materials. Root biomass includes biomass that derives from roots only. We clarified this with new text in Sect. 3.1 (l. 123-125) and a footnote in Table 2.

- **Comment 7**: l. 110: I guess strong agreement is not surprising if the same data are used for different meta-analyses. How much overlap in data used does there exist among the meta-analyses?

   **Response to Comment 7**: This is a great question, and we have to note that we did not set out to do our own meta-analysis, nor to specifically analyze the overlap in data across the meta-analyses we found. We now mention this clearly in Sect. 4.2 (*l*. 398-401). However, one can guess an approximate extent of overlap from the sample size and study year, which we present. The

publication years of the meta-analyses range from 2011 to 2018, and newer studies likely include data that earlier studies could not have included. We think it is important to show that, while every meta-analysis has a unique sample size and time range, there is typically strong agreement among them for any given variable.

- **Comment 8**: l. 234: "humidity affects P deposition". How? I thought most atmospheric deposition of P was in the form of dry deposition.

  **Response to Comment 8**: While some P is dissolved and deposited in rain, mist, and snow, these are not the same as humidity, and the amounts are typically quite small. This phenomenon is not critical for this manuscript, and the statement was somewhat misleading, so we deleted it (Sect. 3.4).

- **Comment 9**: l. 268-272: I don't believe microbial community responses to precipitation changes are as clear as suggested here, and probably strongly depend on timing, intensity and frequency of the precipitation manipulation.

  **Response to Comment 9**: Thank you. We agree that these caveats are important. We added remarks at the conclusion of this section to highlight these dependencies (*l*. 307-311).

**Referee #2: Nameer Baker**

- **Comment 1**: The authors present a useful meta-analysis of meta-analyses on the response of a wide variety of soil factors to increased or decreased precipitation. I believe that the authors have collated published data in a manner that merits publication, but I believe that the results of the study could be significantly improved if a consistent manner to combine and interpret data across meta-analyses could be employed, rather than the method of treating each meta-analysis as an individual unit for comparison. I appreciate that the authors do bring up the sample sizes of each meta-analysis when discussing them and weight the inferences drawn from larger studies more heavily, but I wonder if there is a more effective way to combine the results from the various studies to draw conclusions. Could meta-analyses that presented the same variables have the effect sizes for that variable combined to produce one effect size for the response of the variable to changes in precipitation more generally across studies? This would require knowing the standard deviation of each variable within each meta-analysis, but would make for a much simpler presentation and interpretation of the data, as well as a more valid weighting of the results.

  **Response to Comment 1**: We appreciate this suggestion, and we agree that having one effect size for each variable would greatly simplify the presentation and interpretation. However, we find it challenging to implement because a few 95% CIs are missing, and there are some overlaps of empirical data used among meta-analyses. Therefore, deriving one effect size would not be an accurate calculation. The fact that there are some overlaps of empirical data has been pointed out by the first referee, and for this reason, we included sample size and publication year of each meta-analysis (please see our complete response to the first referee's Comment 7 above; we

included discussion in Sect. 4.2, *l.* 398-401). Furthermore, there are some merits of showing individual meta-analyses; this approach displays the (in)consistency among the meta-analyses' results, and also displays which variables have been more frequently covered (and with greater or lesser sample sizes) compared to other variables.

- **Comment 2**: That is my main request that would require significant alterations to the text and the figures, but I do have some more easily implemented concerns, as well. I wonder if would it be possible to change the abbreviations "IP" and "DP" to something like "up arrow P" and "down arrow P," respectively. This would be easier for the reader to follow in the text, though you might then also want to think about using "W" instead of "P" to refer to precipitation/water to avoid then making it look as though phosphorus content is what is being discussed.

  **Response to Comment 2**: We understand that P could be confusing shorthand for precipitation, but "W" can also cause confusion in the global change community as "W" often refers to warming. We also see IP and DP used in other literature (such as Zhou et al., 2018), and we believe that IP and DP would not cause significant confusion, especially because we clearly define IP and DP in the text and each figure's caption. We very much appreciate this suggestion, but we prefer to keep IP and DP.

- **Comment 3**: Is there also a consistent way to talk about results that had a trend with precipitation, i.e. where something was reduced when precipitation was reduced and increased when precipitation was increased? For instance, saying a variable is positively correlated with precipitation across treatments or negatively correlated with precipitation across treatments? This might also be easier for readers to follow.

  **Response to Comment 3**: While we agree that showing a trend with precipitation could be useful information, it is difficult to make, for example, a graph of x = precipitation change (%) and y = effect size and show the relationship (= trend) because a meta-analytic result could incorporate multiple precipitation change levels. Although we could not quantitatively describe trends across the meta-analyses, qualitative trends are evident for some variables. When qualitative trends can be summarized across treatments, we now describe them (*l.* 179-183, 227-228).

- **Comment 4**: Finally, I wonder if it also might make more sense to group enzyme results with microbial biomass, as they are a microbial response and that way you don't have to spread their discussion out over multiple sections. This is just a suggestion, however.

  **Response to Comment 4**: We initially considered including enzymes in the microbial biomass section. However, as there are respective enzymes for carbon, nitrogen, and phosphorus cycles, we decided to break them into each section. In this way, we were able to summarize enzyme responses in each cycle and relate their responses to other components of the cycle.

- **Comment 5**: 94 – Did you also use Hedge's d for just these variables, or did you then use it for all variables?

**Response to Comment 5**: We used Hedge's d for only these variables.

- **Comment 6**: 122 – This is a place where it would help to be more explicit with your results given that you are saying they are in-line with an expectation, and you can use whatever way you decide to to talk about consistent trends with precipitation (for instance, the response of belowground NPP to both decreasing and increasing precip).

  **Response to Comment 6**: Thank you for your suggestion. We changed the text to be more explicit about our findings, and also included the trend for changes in belowground NPP with precipitation changes (*l*. 133-134).

- **Comment 7**: 127 – If these are general trends across meta-analyses, then how do differences in soil type explain these results more than the differences in nature of the C pool being measured?

  **Response to Comment 7**: Upon reflection, we felt that this section of our manuscript was confusing, and have rewritten it (see lines 139-143). Among other changes, we have removed the reference to soil type here.

- **Comment 8**: 149 – Are there any hypotheses as to why Rh is affected by decreased precip in boreal forest and wetlands, but not in tropical or temperate forests? How about for the effect of increased precipitation in forests and grasslands, but not in wetlands?

  **Response to Comment 8**: We added the following sentences to address this point (*l*. 170-174): "We hesitate to draw strong conclusions from these differences because of the relatively small sample sizes. Zhou et al. (2016), for example, have a sample size of four and five for the tropical and temperate forests, respectively, for DP, and the effects are highly uncertain. The biomes with significant effects – wetlands under DP and grasslands under IP – have higher sample sizes, of 10 and 15, respectively. Biological mechanisms behind these differences can also be hypothesized, such as differences in microbial sensitivity to moisture across systems."

- **Comment 9**: 151 – It is unclear what the conclusion to be drawn from this sentence is.

  **Responses to Comment 9**: This paragraph introduces variability in effect size depending on biomes, methods, and other factors. We added more synthetic paragraph following this sentence that clarifies that these general effects can depend on a variety of factors (*l*. 174-177).

- **Comment 10**: 185 – This is an example of where your presentation and discussion of results would benefit greatly from being able to combine effect sizes across meta-analyses for like variables.

  **Response to Comment 10**: As we described earlier, it is challenging for us to combine effect sizes. Nevertheless, we appreciate this suggestion and leave it for the next project to achieve.

- **Comment 11**: 192 – This is a difficult sentence to parse, I'm not sure how best to remedy it but perhaps something like "However, the product of mineralization and N2 fixation is NH4+, and it increases under DP according to one of three meta-analyses even though fixation could be suppressed."

  **Response to Comment 11**: We modified the sentence accordingly (*l*. 223-225).

- **Comment 12**: 225 – This brings up something that is a bit lacking in discussion of these metaanalyses – are any of them biased or targeted in some fashion, or are they all global? And would a geographic analysis of where all the study sites employed in all of the meta-analyses reveal some obvious blind spots or areas that have been overrepresented in the literature? These would be valuable conclusions to be able to make as a result of your study.

  **Response to Comment 12**: None of the meta-analyses has targeted region/country/biome to conduct their meta-analysis except for Brzostek et al. (2012), meaning that they all include empirical observations from around the world. Yet, the observations are concentrated in the US, Europe, and China, and are sparse in other regions. Brzostek et al. (2012) is US-only, but they include a wide range of ecosystems and biomes. We included this description in the Methods (*l*. 92-95). Since this is a great point to discuss, we added discussion in the Knowledge Gap section as well (*l*. 358-366).

- **Comment 13**: 237 – It may be worth bringing up timescale of studies here for reference relative to P-weathering rates.

  **Response to Comment 13**: We modified the section as suggested (*l*. 258-261).

- **Comment 14**: 255 – It seems that you may be able to draw the conclusion that moisture appears to be generally limiting for microbes in soil.

  **Response to Comment 14**: We added a sentence at the end of the paragraph (*l*. 292-293).

- **Comment 15**: 260 – What direction was this response?

  **Response to Comment 15**: The sentences you pointed to are showing a non-significant effect, so there is no direction. If you meant "Although Blankinship et al. (2011) and Yan et al. (2018) estimated significant effects on the abundance of fungi and F:B ratio (n = 4), ...", both negative and positive effects of IP on the abundance of fungi, and negative effect of DP on F:B ratio. We modified the sentence to clarify the direction (*l*. 297).

- **Comment 16**: 271 – I am not sure that this is the best way to phrase this result, as it appears more that changes in precip don't favor one over the other.

**Response to Comment 16**: We had a similar comment from the first referee as well, and we changed the concluding sentence to emphasize the variability of effects based on the magnitude, duration, and timing of the precipitation treatment (*l.* 307-311).

- **Comment 17**: 281 – In what direction could the ratio be altered?

  **Response to Comment 17**: As MBC:MBN increased with IP, soil microbial biomass C:N:P could also be increased to have more weight on carbon. We modified the sentence to clarify this point (*l.* 323-324).

- **Comment 18**: 283 – How would the mycorrhizal symbiosis change the dynamics? A bit more detail would be useful to the reader.

  **Response to Comment 18**: Strong mycorrhizal symbiosis might be able to help a plant with nutrient uptake under DP and help maintain the soil N:P ratio. We added a sentence to explain this (*l.* 326-327).

- **Comment 19**: 287-292 – This section feels sparse, and would be well-served to also bring up ecosystems or geographic regions that have been under- or over-sampled, as mentioned in a previous comment. Also, what about the paucity of studies that have measured bacterial:fungal biomass responses to increased precipitation?

  **Response to Comment 19**: It is a great point to include the geographic differences in observations, as well as the paucity of studies in bacterial:fungal biomass responses. We improved the entire section to reflect this suggestion (Sect. 4.1).

- **Comment 20**: 301 – Some more discussion of what this blind spot in terms of N-process rates means for inferring conclusions about the N-cycle in soil would be useful to the reader to understand why this is valuable fruit to pursue.

  **Response to Comment 20**: We had a similar comment from the first referee as well, and we agree that we needed to elaborate on the importance of these nitrogen process rates variables. We improved the section to reflect your suggestion (Sect. 4.1, *l.* 349-356).

- **Comment 21**: 317 – Do you have any suggestions as to what types of data formatting / archiving you ran into that was helpful or a hindrance? You have an opportunity to say some things from this pulpit, take advantage!

  **Response to Comment 21**: Great suggestion. We added discussion in Sect. 4.2.

- **Comment 22**: 322 – I'm not sure this statement is quite true given the response of microbial biomass and the crude measures of microbial community assayed – it is fair to say that the ratio of fungi to bacterial biomass is insensitive, but that is not the same as the community being resistant.

**Response to Comment 22**: We agree with this point, and modified the section based on your comment (Sect. 5).

- **Comment 23**: Figures – Could you bold the symbols used to indicate the direction of the effect to make them stand out more? Also, if you are not going to use the raindrops to denote precip effects on each flow-figure then don't use it on any of them.

  **Response to Comment 23**: We bolded the symbols used in Figures 1 and 3, and removed the raindrops in Figure 1.

**References**

[revised manuscript text omitted]

a. Belowground biomass was measured by drying soil cores (Wu et al., 2011), and thus includes roots and other plant- and animal-derived materials. Root biomass includes biomass that derives from roots only.

685 b. C-acquisition enzymes are β-1,4-glucosidase and β-D-cellobiohydrolase, N-acquisition enzymes are β-1,4-N-acetyl-glucosaminnidase, leucine amino peptidase, and urease, and the P-acquisition enzyme is acid phosphatase (Xiao et al., 2018).

[Figure]

(a)

(b)

(c) Decreased precipitation

(d) Increased precipitation

[revised manuscript text omitted]

---

## Author Response (AR3)

Dear Dr Kees Jan van Groenigen,

We would like to thank you for accepting our manuscript for publication in Biogeosciences. We appreciate your timely responses and thoughtful comments in this difficult time. We hope you and your family stay safe and healthy.

Best regards,

Akane O Abbasi et al.